# Benefits of Harmonicity for Hearing in Noise Are Limited to Detection and Pitch-Related Discrimination Tasks

**DOI:** 10.3390/biology12121522

**Published:** 2023-12-13

**Authors:** Neha Rajappa, Daniel R. Guest, Andrew J. Oxenham

**Affiliations:** 1Department of Psychology, University of Minnesota, Minneapolis, MN 55455, USA; rajap013@umn.edu; 2Department of Biomedical Engineering, University of Rochester, Rochester, NY 14627, USA; daniel_guest@urmc.rochester.edu

**Keywords:** amplitude modulation, fundamental frequency, frequency modulation, harmonicity, harmonic benefit, pitch discrimination, musicianship

## Abstract

**Simple Summary:**

Harmonic tones are known to be more detectable in noise than inharmonic tones, providing an ecological advantage to periodic sounds, such as voiced speech and many animal vocalizations, over aperiodic sounds. Little is known about the extent to which this harmonic advantage extends to other auditory tasks beyond simple detection. Here, we tested young adults with and without musical training on the detection of harmonic and inharmonic tones in noise, as well as on other tasks that are important for auditory perception, including pitch discrimination, frequency-modulation detection, and amplitude-modulation detection. Consistent with earlier results, detection in noise was superior for harmonic over inharmonic tones, and an additional benefit beyond detection was found for pitch discrimination. However, neither amplitude- nor frequency-modulation detection was improved in harmonic conditions, beyond the predicted effects of audibility. Listeners with musical training performed better in pitch-related tasks (pitch discrimination and frequency-modulation detection), but evidence for a musician benefit of harmonicity was weak. The results show that harmonicity aids tone detection and pitch perception in noise, but that the harmonicity advantage does not extend to other important auditory domains, such as modulation detection, beyond the improvements accounted for by greater audibility in noise.

**Abstract:**

Harmonic complex tones are easier to detect in noise than inharmonic complex tones, providing a potential perceptual advantage in complex auditory environments. Here, we explored whether the harmonic advantage extends to other auditory tasks that are important for navigating a noisy auditory environment, such as amplitude- and frequency-modulation detection. Sixty young normal-hearing listeners were tested, divided into two equal groups with and without musical training. Consistent with earlier studies, harmonic tones were easier to detect in noise than inharmonic tones, with a signal-to-noise ratio (SNR) advantage of about 2.5 dB, and the pitch discrimination of the harmonic tones was more accurate than that of inharmonic tones, even after differences in audibility were accounted for. In contrast, neither amplitude- nor frequency-modulation detection was superior with harmonic tones once differences in audibility were accounted for. Musical training was associated with better performance only in pitch-discrimination and frequency-modulation-detection tasks. The results confirm a detection and pitch-perception advantage for harmonic tones but reveal that the harmonic benefits do not extend to suprathreshold tasks that do not rely on extracting the fundamental frequency. A general theory is proposed that may account for the effects of both noise and memory on pitch-discrimination differences between harmonic and inharmonic tones.

## 1. Introduction

Periodic or harmonic sounds are characterized by a regular (harmonic) spectrum and are ubiquitous in our environment in the form of human voices, animal vocalizations, and musical instrument sounds. These sounds typically elicit a percept of pitch corresponding to the repetition rate or fundamental frequency (F0) of the sound [1,2]. Pitch perception plays an important role in understanding speech (intonation, stress, prosody), enjoying music (melody, harmony), and segregating competing sounds in complex environments. Because of the importance and ubiquity of harmonic sounds and their pitch, the human auditory system may be adapted to preferentially process them through pitch- or harmonic-sensitive neurons and brain regions [3,4,5,6,7].

In an important perceptual study, Hafter and Saberi reported that humans are better able to detect a simultaneous combination of pure tones in noise when they are in a harmonic relation than when they are inharmonic [8]. The authors interpreted this finding as evidence for a higher level of analysis than simply the tonotopic (spectral) representation found along the cochlear partition or in the auditory nerve, potentially based on neural representations of pitch. The finding of a detection advantage in noise for harmonic tones was recently replicated and extended to show that the advantage generalized to improved pitch discrimination of the tones in noise, even after accounting for the enhanced audibility of harmonic tones [9].

A harmonic advantage for detecting complex tones in noise may have important implications for understanding speech in noise, or for detecting animal vocalizations within complex acoustic backgrounds. However, it remains unclear to what extent this harmonic advantage generalizes to tasks beyond detection and pitch discrimination, such as the detection of amplitude and frequency modulations, both of which are critical for speech understanding and general vocal communication. Because frequency modulation (FM) detection and F0 discrimination are generally believed to rely on similar mechanisms [10,11] at least for slow modulation rates [12], it seems plausible that a harmonic advantage would also be observed for FM detection. In contrast, the detection of amplitude modulation (AM) does not rely on extracting the pitch or F0 of a stimulus and so may show no benefit of harmonicity beyond that provided by improved audibility. The aim of this study was, therefore, to determine whether the harmonic advantage extends to the FM and AM detection, both of which are important for detecting and identifying ecologically relevant sounds in noise.

Musical training has often been associated with better performance in pitch-related tasks but less so in tasks not involving pitch [13,14]. We therefore expected that musical training may be associated with better FM, but not AM, detection. In terms of harmonicity, musicians may be more sensitive to it than non-musicians when judging consonance and dissonance [15], but they do not appear to exhibit a greater harmonic advantage than non-musicians for either tone detection or pitch discrimination [9]. To test for any effects of musicianship in our study, we selected participants to be evenly divided between those who had extensive musical training and those who did not. All listeners were tested for their tone-detection and F0-discrimination thresholds, as well as for their AM- and FM-detection thresholds using both harmonic and inharmonic complex tones in noise. Our initial hypotheses were that (1) harmonicity would aid FM detection, but not AM detection, in complex tones beyond the expected improvement in audibility (similar to that found for F0 discrimination), and that (2) musicianship would be associated with better performance in pitch-related tasks (F0 discrimination and FM detection) but not the others (tone detection and AM detection).

## 2. Materials and Methods

### 2.1. Participants and Procedure

A total of 60 participants were tested (34 female, 26 male, age range 18–24 years), divided equally into two groups of musicians and non-musicians. Musicians (19 female, 11 male) were defined as having at least 10 years of musical training and being currently active in practice at the time of the experiment. Non-musicians (15 female, 15 male) had less than 2 years of musical training and were not actively playing a musical instrument at the time of the experiment. Each participant underwent audiometric screening, which included measuring auditory thresholds at octave frequencies from 250 to 8000 Hz. All participants had audiometric thresholds < 20 dB hearing level (HL) in both ears at all tested frequencies and reported no history of hearing disorders. A very small number of participants did not complete all of the experiments; for each experiment described below, the rate of attrition was noted. Participants provided written informed consent and received compensation in the form of hourly payment or course credit for their participation. The experimental protocols were approved by the Institutional Review Board of the University of Minnesota. Experiments were conducted in sound-attenuating booths, with the AFC toolbox within MATLAB used to present the stimuli and collect responses [16]. Stimuli were presented diotically from an E22 soundcard (LynxStudio, Costa Mesa, CA, USA) at a 24-bit resolution and a sampling rate of 48 kHz via HD650 circumaural headphones.

### 2.2. Masking Noise

All experiments employed threshold-equalizing noise (TEN) as a masking noise [17]. TEN has the advantage of providing roughly equal detectability for pure tones, independent of the tone frequency. In all cases, TEN was generated at a level of 50 dB SPL within the estimated equivalent rectangular bandwidth (ERB) of the human auditory filter centered at 1 kHz [18]. It was then zero-phase-lowpass-filtered with a 4th-order Butterworth lowpass filter with a cutoff frequency of 8 kHz to eliminate unnecessary high-frequency content from the noise for comfort. On each trial, TEN was gated on with a 50-ms raised-cosine ramp 200 ms before the beginning of the test stimuli and gated off with a 50-ms raised-cosine ramp 200 ms after the end of the test stimuli.

### 2.3. Detection in Noise

We first measured detection thresholds for harmonic and inharmonic complex tones embedded in TEN using an adaptive procedure. All 60 participants completed this experiment. The purpose of these measurements was to test for a difference in audibility between harmonic and inharmonic complex tones in noise and to ensure that our stimuli would be audible at all tested signal-to-noise ratio (SNR) values in the other tasks.

Each trial of the procedure was divided into two 1-s intervals that were separated by 200 ms and marked by lights on the virtual response box. One of the two intervals, selected at random on each trial, contained the 1-s target tone (including 50-ms raised-cosine onset and offset ramps). Listeners were tasked with identifying which interval contained the target. Feedback was provided immediately after the response on each trial. The target was composed of the first 15 harmonics of a nominal 250 Hz F0, synthesized in random phase, and then zero-phase-filtered with an 8th-order Butterworth bandpass filter with cutoffs at 750 and 3000 Hz (3× and 12× F0). The actual F0 varied from trial to trial uniformly over a range of ±10% of the nominal F0. Targets were either harmonic, in which case component frequencies were exact integer multiples of the F0, or they were inharmonic, in which case the frequency of each component was independently jittered around the corresponding harmonic frequency according to a uniform distribution spanning from −F0/2 to +F0/2 (i.e., a ±50% jitter), newly selected for each trial. The SNR was specified in terms of the level per component in the passband of the complex tone, relative to the 50 dB SPL TEN level per ERB. The range of possible jitter values was constrained by iteratively generating sets of values and rejecting those that failed to meet two criteria until 10,000 accepted sets of values were generated. The first criterion was that jittered frequency values could not result in two components falling within 0.05 F0 of each other. This criterion was intended to ensure that strong beats from very closely spaced components would not be present in the stimuli. The second criterion was more complex. For each candidate set of jittered frequency values, we synthesized the complex-tone stimulus without noise and then computed the autocorrelation function of the resulting stimulus waveform. Peaks in the autocorrelation function were identified using the find_peaks function in the Scipy Python 3 package with the distance parameter set to 1 ms and all other parameters at default values. If any identified peak fell within ±3 bins of the bin of the lag axis closest to the F0 lag (1/250 Hz = 4 ms), the criterion was deemed not met, and the set was rejected. This criterion was intended to ensure that a strong periodicity at F0 was not present in the resulting inharmonic complex tones by chance. The resulting sets were then randomly sampled to generate jittered frequency values on each trial.

We used a two-interval two-alternative forced-choice (2I2AFC) task with a 3-down 1-up (3D1U) adaptive staircase procedure to track the 79.4% point on the psychometric function [19]. Each run began at an SNR value of 5 dB. The initial step size of the run was 3 dB. After two reversals, the step size was reduced to 1.5 dB. After two more reversals, the step size was reduced to 0.75 dB, and the run continued for another six reversals. The threshold for each such run was defined as the mean SNR at the last six reversal points. The two conditions, harmonic and inharmonic, were each tested three times. The order of presentation was randomized for each repetition and each listener. The final threshold for each listener and condition was defined as the mean threshold across the three repetitions.

### 2.4. Fundamental Frequency Discrimination

Next, F0 discrimination thresholds were measured in a 2I2AFC task with a 3D1U adaptive procedure. All 60 participants completed this experiment. Stimuli were synthesized and presented as described above in the detection task, except that both intervals contained a complex tone. One of the tones had a lower F0 and one a higher F0, with the order selected randomly on each trial. The F0s were geometrically centered around the F0 selected on each trial from the F0 rove distribution (see above). For the inharmonic condition, the jitter values varied between trials but were held constant within each trial (across the two intervals). Listeners were tasked with indicating which interval had the higher pitch. Feedback was provided immediately after the response on each trial. The initial value of ΔF0 (the difference in F0 between the tones in the two intervals) was 10% of the lower F0. Initially, this value was adjusted (increased or decreased) by a factor of 2. After two reversals, the step size was reduced to factor of 1.41 (2^1/2^). After two more reversals, the step size was reduced to a factor of 1.19 (2^1/4^) and the procedure continued for six more reversals. Threshold was defined as the geometric mean of the values of ΔF0 at the last six reversal points. Both the harmonic and inharmonic tones were tested at three SNRs (−2.5 dB, 0 dB, 2.5 dB) as well as at a level of 50 dB SPL per component in quiet, to yield a total of eight conditions. These conditions were tested in random order for each of the two repetitions and each listener. For each condition and listener, the overall threshold was defined as the geometric mean threshold across the two runs. The adaptive procedure had a maximum allowable ΔF0 of 100%. If the tracking procedure called for a ΔF0 value that exceeded the maximum on six trials within a given run, the procedure was terminated early, and the threshold for that run was defined as 100%.

### 2.5. Frequency (F0) Modulation Discrimination

Next, FM detection thresholds were measured in a 2I2AFC task with a 3D1U adaptive procedure. Three of the 60 participants did not complete this experiment. Stimuli were synthesized as described above in the F0 task, except that each interval had the same average F0 and one interval included frequency/F0 modulation while the other did not (order selected randomly on each trial). Listeners were tasked with indicating which interval contained the modulation or was changing. Feedback was provided after each trial. The modulated tones were synthesized by applying sinusoidal FM to each individual component before summation and filtering, according to the equation:(1)$$x_{n}\left(t\right)=sin(2\pi {nf}_{0}t+\phi +\left(\frac{\delta_{f}}{f_{m}}\right)\sin(2\pi f_{m}t+\Phi_{m})),$$where $x_{n}$ is the time waveform of the nth component of the complex tone, $f_{0}$ is the F0, n is the harmonic number of the component, $f_{m}$ is the modulator frequency, ϕ is the carrier phase, $\delta_{f}$ is the frequency excursion from the carrier frequency, and $\Phi_{m}$ is the modulator phase. The modulator frequency was always set to 2 Hz, and a single modulator phase for all modulators was selected randomly within the range [0,2π] from trial to trial. The initial frequency excursion, $\delta_{f}$, was set to 3.16% of the carrier frequency. This value was initially varied by a factor of 2. After the first two reversals, $\delta_{f}$ was varied by a factor of 1.41. After two more reversals, the value was adjusted by a factor of 1.19 for the final six reversals of the run. Threshold was defined as the geometric mean of the $\delta_{f}$ value at the last six reversal points. As for F0 discrimination, harmonic and inharmonic conditions were tested at three SNRs (−2.5 dB, 0 dB, 2.5 dB) and in quiet (with the tones set to 50 dB SPL per component) to yield eight unique conditions, with two threshold estimates obtained for each condition and listener. As before, the conditions were presented in random order for each listener and repetition. The adaptive procedure had a maximum allowable peak-to-peak frequency excursion in percent of the carrier frequency of 100%. If the tracking procedure called for a carrier frequency value that exceeded the maximum on six trials within a given run, the procedure was terminated early, and the threshold for that run was defined as 100%. Each listener’s threshold was defined as the geometric mean across the two runs.

### 2.6. Amplitude Modulation Discrimination

Finally, AM detection thresholds were measured in a 2I2AFC task with a 3D1U adaptive procedure. Three of the 60 participants did not complete this experiment. Stimuli were synthesized as described above in the FM task, except that each interval within a trial had the same (constant) F0, and one interval included amplitude modulation while the other did not (order selected randomly on each trial). Listeners were tasked with indicating which interval contained the modulation or was changing. Amplitude modulation was applied according to the equation:(2)$$y\left(t\right)=(1+m\sin(2\pi ft))x(t),$$
where *x*(*t*) is the unmodulated complex tone, *f* is the modulator frequency, and *m* is the modulation index. The modulator frequency was set to 2 Hz. The tracked variable of the procedure was 20log_10_(*m*). The tracking variable began at a value of −10 dB and was adjusted with an initial step size of 3 dB. After two reversals, this step size was reduced to 1.5 dB, and after two more reversals, the step size was reduced further to 0.75 dB. After 6 reversals at this final step size, the threshold was taken as the mean of the tracked values at the last 6 reversal points. As for F0 discrimination, two levels of harmonicity (harmonic and inharmonic) and four levels of SNR (−2.5 dB, 0 dB, 2.5 dB, and quiet) were tested in all combinations to yield 8 unique conditions, for which two thresholds were measured in random order in each listener. The adaptive procedure had a maximum allowable modulation depth of 0 dB. If the tracking procedure called for a modulation depth value that exceeded the maximum on six trials within a given run, the procedure was terminated early, and the threshold for that run was defined as 0 dB. Each listener’s threshold was defined as the arithmetic mean across the two runs.

### 2.7. Statistical Analysis

Thresholds were analyzed by fitting linear mixed-effects models to the data, using harmonicity (harmonic, inharmonic), SNR (−2.5, 0, 2.5 dB, or quiet), and musicianship (musician, non-musician) as categorical fixed-effects predictors. All possible interactions between predictors were included. Random effects included listener intercepts and listener slopes for SNR and harmonicity. Models were fitted with default optimization settings using the lme4 package in the R 4.3.0 programming language and then analyzed using F-tests in a Type II analysis of variance (ANOVA) via the car package with the Kenward–Roger approximation [20,21]. When necessary, the ANOVA was supplemented by F-tests of linear contrasts over model coefficients to test specific effects or post hoc observations. For each experiment, these tests were corrected for multiple comparisons using the Holm–Bonferroni method [22].

To determine whether differences in audibility between harmonic and inharmonic stimuli could explain any significant effects of harmonicity, we fit additional models to subsets of the data from each experiment. Specifically, we included data from −2.5 and 0 dB SNRs for the harmonic conditions and from 0 and +2.5 dB SNRs for the inharmonic conditions. In these alternative models, instead of using SNR as a fixed-effect predictor, we used a new categorical variable with two levels. Data from the −2.5 dB SNR harmonic condition and the 0 dB SNR inharmonic condition were assigned to the first level, while data from the 0 dB SNR harmonic condition and 2.5 dB inharmonic condition were assigned to the second level. This approach reflected the fact that inharmonic detection thresholds were approximately 2.5 dB worse than the harmonic detection thresholds (see Section 3.1). Thus, these alternative models quantified the effect of harmonicity at a constant dB sensation level (SL) rather than at a constant SNR. We then analyzed the models using the same ANOVA procedure described above. If the effects of harmonicity or its interactions were no longer significant, we interpreted this as evidence that significant effects in the original model were driven by differences in audibility. As is traditionally the case, all threshold values were log-transformed, either through the use of dB (tone detection, AM detection) or by using log-transformed values of F0 differences. These transforms provide more uniform variance across conditions and are theoretically justified by the proportionality of sensitivity (*d*’) and the independent variables (tone intensity, modulation depth, and frequency difference) [23].

## 3. Results

### 3.1. Detection in Noise

Individual and mean detection thresholds for harmonic and inharmonic tones in noise are shown in Figure 1, separately for the musicians and non-musicians. The ANOVA revealed a significant main effect of harmonicity (F_1, 59_ = 149, *p* < 0.001), indicating that thresholds differed significantly for harmonic and inharmonic stimuli, with thresholds for the harmonic tones approximately 2.5 dB lower (better) on average than those for the inharmonic tones. Neither the main effect of musicianship (F_1, 59_ = 2.19, *p* = 0.144) nor its interaction with harmonicity (F_1, 59_ = 1.34, *p* = 0.253) was significant, suggesting that musicians and non-musicians are equally adept at detecting tones in noise, with both showing the same detection advantage for harmonic over inharmonic tones in noise.

### 3.2. Fundamental Frequency Discrimination

The individual and mean thresholds for F0 discrimination are shown in Figure 2. The ANOVA using log-transformed discrimination thresholds as the dependent variable revealed significant main effects of SNR (F_3, 56_ = 134, *p* < 0.001), harmonicity (F_1, 59_ = 242, *p* < 0.001), and musicianship (F_1, 59_ = 14.7, *p* = 0.001). A significant two-way interaction was observed between SNR and harmonicity (F_3, 657_ = 62.9, *p* < 0.001), SNR and musicianship (F_3, 57_ = 3.19, *p* = 0.031), and harmonicity and musicianship (F_1, 59_ = 8.72, *p* = 0.005), along with a significant three-way interaction (F_3, 657_ = 6.87, *p* < 0.001). Other terms did not attain significance.

A harmonic advantage was observed in noise, with better average F0 discrimination thresholds for harmonic than inharmonic tones at −2.5, 0, and 2.5 dB SNRs (average harmonic threshold = 3.2%, average inharmonic threshold = 23.4%; all pairwise comparisons F_1, 58_ > 62.1, *p* < 0.001) but not in the quiet condition (average harmonic threshold = 1.4%, average inharmonic threshold = 1.6%; F_1, 58_ = 1.28, *p* = 0.263). This difference in noise, which does not occur in quiet, appears to be responsible for the significant interaction between harmonicity and SNR. Overall, musicians’ thresholds were lower than those of non-musicians, as indicated by the main effect of musicianship (average musician threshold = 3.3%, average non-musician threshold = 9.2%). The interaction between harmonicity and musicianship seems to be explained by the apparently larger difference in thresholds between musicians and non-musicians in harmonic than inharmonic conditions in noise (ratio of average non-musician/musician inharmonic thresholds = 1.66, ratio of average non-musician/musician harmonic thresholds = 3.95; significant at SNR = −2.5 dB, F_1, 271_ = 16.7, *p* < 0.001; significant at SNR = 0 dB, F _1, 271_ = 11.9, *p* = 0.002). However, this observation should be qualified by noting that many thresholds in the inharmonic conditions in noise were unmeasurable and were therefore set to 100%, potentially underestimating the difference between musicians and non-musicians in the inharmonic conditions. These ceiling effects may also explain the significant interaction between musicianship and SNR. Also of note were the large individual differences in performance within each group.

It is possible that some of the differences in performance between harmonic and inharmonic tones in noise could be attributed to differences in the audibility of the tones in noise. However, given that the average difference in detection threshold was about 2.5 dB (Figure 1), it is clear from Figure 2 (note inset) that audibility alone cannot account for differences in F0 discrimination thresholds. To formally test this assertion, we re-analyzed a subset of the data with an additional ANOVA that grouped data based on the average SL at which they were collected, rather than the SNR (see Methods). This ANOVA revealed that the main effect of harmonicity was still significant (F_1, 59_ = 113, *p* < 0.001). Additionally, the significant interaction between harmonicity and musicians remained significant (F_1, 59_ = 6.37, *p* = 0.014). This analysis confirms that the effects of harmonicity on F0 discrimination measured here extended beyond the expected effects due to differences in audibility.

### 3.3. Frequency Modulation Detection

The individual and mean thresholds for FM detection are shown in Figure 3. The ANOVA using log-transformed FM detection thresholds (peak-to-peak frequency deviation as a proportion of the carrier frequency) as the dependent variable revealed significant main effects of SNR (F_3, 54_ = 219, *p* < 0.001), harmonicity (F_1, 56_ = 21.9, *p* < 0.001), and musicianship (F_1, 56_ = 8.80, *p* = 0.004). The interaction between SNR and harmonicity was significant (F_3, 619_ = 29.7, *p* < 0.001); however, neither the remaining two-way interactions, between SNR and musicianship (F_3, 54_ = 0.528, *p* = 0.665) and between harmonicity and musicianship (F_1, 56_ = 0.585, *p* = 0.448), nor the three-way interaction (F_3, 619_ = 0.659, *p* = 0.578) were significant. The main effects of SNR and musicianship reflect the improvement in performance with increasing SNR and the better performance (lower thresholds) of the musicians, relative to the non-musicians. The main effect of harmonicity and the SNR-by-harmonicity interaction seem to be driven by the improved performance with harmonic over inharmonic tones at the lowest SNR of −2.5 dB.

In contrast to the F0 discrimination results, the effect of harmonicity on FM detection was small and appeared to be mostly accounted for by differences in audibility (inset, Figure 3). As for F0 detection, we sought to quantify this effect by re-analyzing a subset of the data in terms of SL instead of in terms of SNR. The effect of harmonicity remained significant (F_1, 56_ = 17.5, *p* < 0.001) when analyzed in this way, but visual inspection revealed that this was because FM detection thresholds were actually better for inharmonic stimuli than for harmonic stimuli at the same SL (see inset, Figure 3). Thus, there is no evidence that harmonicity improves FM detection beyond the effects of improved audibility.

### 3.4. Amplitude Modulation Detection

The individual and mean thresholds for AM detection are shown in Figure 4. The ANOVA with AM detection thresholds as the dependent variable revealed a significant main effect of SNR (F_3, 54_ = 169, *p* < 0.001) and harmonicity (F_1, 56_ = 51.0, *p* < 0.001). There was no main effect of musicianship (F_1,56_ = 0.209, *p* = 0.649). The two-way interaction between SNR and harmonicity was significant (F_3, 628_ = 25.8, *p* < 0.001), as was the interaction between SNR and musicianship (F_3, 54_ = 5.27, *p* = 0.003). Neither the interaction between harmonicity and musicianship (F_1, 56_ = 0.439, *p* = 0.510) nor the three-way interaction was significant (F_3, 629_ = 1.38, *p* = 0.246).

Overall, the pattern of results was similar to that found for FM detection. In line with the main effect of harmonicity, performance in the inharmonic condition was worse than in the harmonic condition at all SNRs, excluding in quiet (all F_1, 34–45_ > 34.2, all *p* < 0.001). The significant interaction between SNR and musicianship reflects the appearance of somewhat lower (better) thresholds for the musicians than non-musicians at a 2.5 dB SNR and in quiet; post hoc testing revealed that these differences were marginally significant after correction (2.5 dB, F_1, 56_ = 6.02, *p* = 0.052; quiet, F_3, 629_ = 6.59, *p* = 0.052). As for FM detection, we sought to determine if the effect of harmonicity in noise could be attributed to differences in audibility. A follow-up ANOVA applied to a subset of the data showed that the effect of harmonicity was no longer significant when data were analyzed in terms of SL instead of SNR (F_1, 56_ = 1.06, *p* = 0.308) (see inset, Figure 4). In other words, differences in AM detection for harmonic and inharmonic stimuli could be attributed to differences in audibility.

## 4. Discussion

In the present experiments, we measured tone detection, F0 discrimination, FM detection, and AM detection for harmonic and inharmonic complex tones in musicians and non-musicians. We confirmed a benefit of harmonicity of about 2.5 dB for the tone detection threshold [8]. We also confirmed a harmonic benefit for F0 discrimination in noise that was greater than could be accounted for simply by the differences in audibility [9]. In contrast, for AM and FM detection, the harmonic benefits observed could be entirely attributed to differences in audibility. Our first hypothesis (that harmonicity would aid FM but not AM detection) was, therefore, only partially supported. Musicians performed better than non-musicians in the tasks involving detecting changes in pitch or frequency (F0 discrimination and FM detection) but not in the others (tone detection and AM detection). This selective musician benefit is consistent with multiple prior studies showing that musicians demonstrate superior pitch discrimination skills compared to non-musicians [14,24,25]. There was some indication that musicians made more use of harmonicity in F0 discrimination than non-musicians. However, that finding may be due to the large number of non-musicians whose F0 discrimination thresholds could not be measured in the inharmonic conditions at the lowest SNRs, leading to a potential underestimation of F0 discrimination threshold differences and, hence, a possibly spurious interaction effect due to ceiling effects. With the exception of this interaction, our results are consistent with those of McPherson et al. [9], who also found an effect of harmonicity on tone detection and F0 discrimination but no interactions with musicianship. However, our finding of no benefit of harmonicity beyond audibility for both AM and FM detection is novel.

Why would F0 discrimination, but not slow-rate FM detection, show a harmonic advantage? This result seems puzzling, as both tasks have been hypothesized to reflect the same underlying mechanisms [10,11]. One explanation relates to the idea that the audibility (and salience) of individual components within each complex tone in noise varies randomly from trial to trial. For FM detection, only one component is sufficient to determine whether an interval contains FM or not. In contrast, for the F0 discrimination, components must be compared between intervals within a trial to determine which is higher. In harmonic conditions, one near-threshold component can be sufficient to convey the underlying F0 [26]. Thus, even if different harmonics of the F0 are more audible between the two intervals, listeners may still accurately hear the F0 rise or fall across the two-interval trial. In contrast, for the inharmonic tones, there is no unambiguous F0, and so listeners must rely on the rise or fall in frequency of individual components between the two trials. If different components are more audible or salient between the two trials, then listeners would not be able to reliably complete the task, as we observed.

This explanation can be extended to provide a more general theory of why increasing the interstimulus interval between trials has been found to affect inharmonic pitch discrimination more than harmonic pitch discrimination [27]. In their study, McPherson and McDermott speculated that the pitch discrimination of inharmonic tones was more dependent on interstimulus interval because the memory of the inharmonic sound decayed more rapidly than the memory of the global pitch associated with the harmonic sound. It is not clear how such an account could also explain our finding of harmonic benefits to F0 discrimination at low SNRs. In contrast, our explanation based on the trial-by-trial variability of the salience of individual components provides a qualitative account of both findings. In the memory task of McPherson and McDermott, listeners may follow one or more components in the inharmonic tone and note whether that component rose or fell in frequency from the first to the second interval. As the gap between intervals becomes longer, listeners may not be able to follow the direction of the pitch change of a specific component; indeed, the component(s) that (randomly) garner the most attention in the first interval may not be the same components that garner the most attention in the second interval, leading to poor performance. Similarly, at very low SNRs (as in our experiment), the salience of individual components likely varies based on noise variability and random interactions between the noise and each component [28]. In both cases (long gaps and low SNRs), reliance on individual components results in poorer performance for inharmonic than harmonic tone complexes.

Aside from not affecting FM perception, inharmonicity also did not affect AM detection thresholds. Given that inharmonicity results in irregular and unpredictable amplitude fluctuations (or beats) between adjacent components, a reasonable prediction would have been that these amplitude fluctuations may have led to modulation interference or masking [29] and, hence, poorer AM detection thresholds. The reason that no such masking was observed is likely because the target modulation rate of 2 Hz was very low, relative to the beat frequencies, which had an expected rate of 250 Hz, and no rates were below 12.5 Hz, due to the constraint that no components could be closer than 0.05F0 (or 12.5 Hz) apart. Because AM masking is frequency selective [29,30], it is expected that masker modulation at least two octaves higher than the target modulation should not have a substantial influence on AM detection thresholds.

Finally, what are the neural substrates that might account for the harmonic benefit? Hafter and Saberi’s initial study hypothesized the existence of a higher level of analysis that could operate on, for instance, pitch rather than the individual tones within a complex [8]. Evidence for pitch or harmonicity sensitivity and selectivity has been found in human and non-human-primate studies of auditory cortex [4,5,6,7,31]. Such cortical representations may provide the substrate of the higher-level analyses required to provide a harmonic advantage in noise.

## 5. Conclusions

F0 discrimination, but not FM and AM detection, showed a significant harmonic advantage in noise after accounting for differences in the audibility of harmonic and inharmonic tones. Musicianship was associated with better performance in pitch-related tasks but not in tone detection or AM detection. Overall, the effects of harmonicity are consistent with neural specializations for harmonic sounds in humans that provide specific behavioral advantages for ecologically relevant sound detection and discrimination in noise.

## Figures and Tables

**Figure 1 biology-12-01522-f001:**
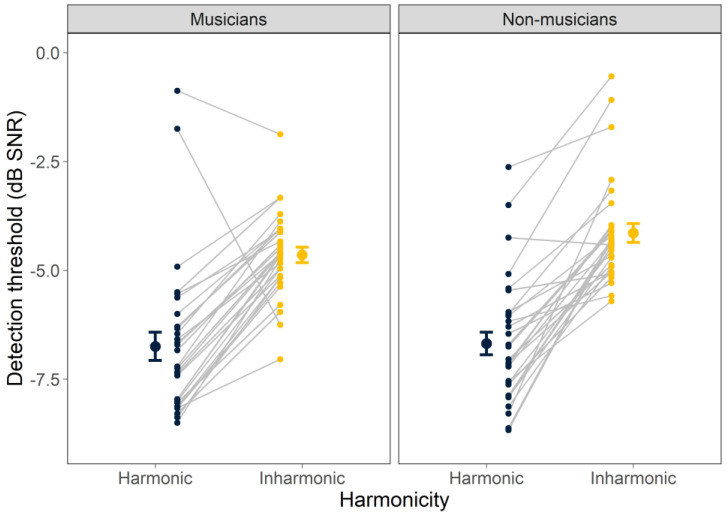
Detection thresholds for harmonic and inharmonic complex tones in noise. Results are separated by musicians (**left**) and non-musicians (**right**). Color indicates harmonicity. Individual thresholds are shown as smaller symbols; larger symbols represent mean thresholds. Error bars indicate ±1 standard error of the mean.

**Figure 2 biology-12-01522-f002:**
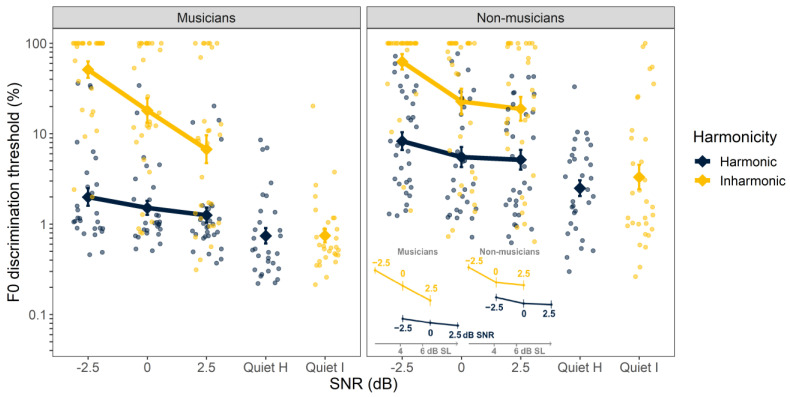
Fundamental frequency (F0) discrimination thresholds as a function of signal-to-noise ratio (SNR). Results are separated by musicians (**left**) and non-musicians (**right**). Harmonic and inharmonic thresholds in quiet are plotted with a horizontal offset for clarity. Color indicates harmonicity. Larger symbols represent the mean across participants. Smaller symbols repreent thresholds from individual listeners. The inset figure replots the mean data with SL on the abscissa to illustrate that the harmonicity effect remains even after differences in audibility are accounted for. Error bars indicate ±1 standard error of the mean. Inset *x*-axis values indicate approximate average dB SL values.

**Figure 3 biology-12-01522-f003:**
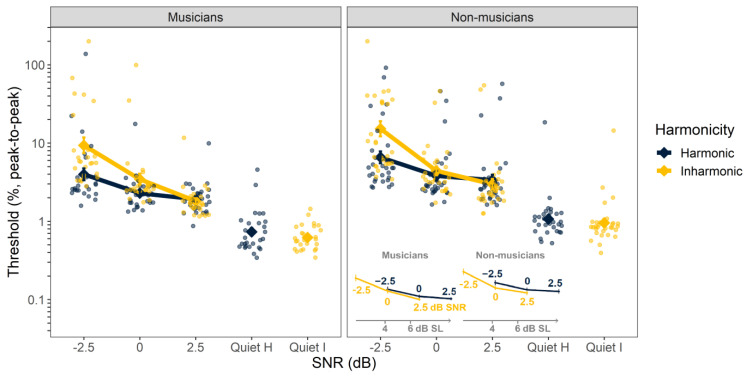
Frequency modulation detection thresholds as a function of signal-to-noise ratio (SNR). Results are separated by musicians (**left**) and non-musicians (**right**). Harmonic and inharmonic thresholds in quiet are plotted with a horizontal offset for clarity. Color indicates harmonicity. Larger symbols represent the mean across participants. Smaller symbols represent thresholds from individual listeners. Error bars indicate ±1 standard error of the mean. The inset figure replots the mean data with SL on the abcissa to illustrate that the harmonicity advantage is no longer present after differences in audibility are accounted for. Inset *x*-axis values indicate approximate average dB SL values.

**Figure 4 biology-12-01522-f004:**
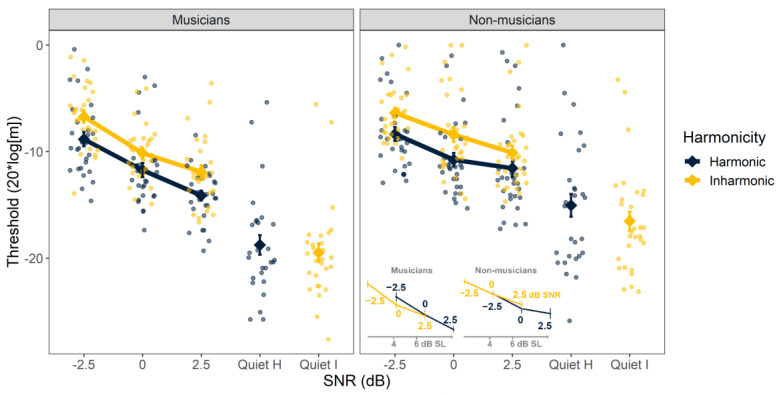
Amplitude modulation detection thresholds as a function of signal-to-noise ratio (SNR). Results are separated by musicians (**left**) and non-musicians (**right**). Harmonic and inharmonic thresholds in quiet are plotted with a horizontal offset for clarity. Color indicates harmonicity. Larger symbols represent the mean across participants. Smaller symbols represent thresholds from individual listeners. Error bars indicate ±1 standard error of the mean. The inset figure replots the mean data with SL on the *x*-axis to illustrate that the harmonicity advantage is no longer present once differences in audibility are accounted for. Inset *x*-axis values indicate approximate average dB SL values.

## Data Availability

Data are available upon request.

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
