# Peer review of "Benefits of Harmonicity for Hearing in Noise Are Limited to Detection and Pitch-Related Discrimination Tasks"

_biology, 2023, doi:10.3390/biology12121522_

Round 1
Reviewer 1 Report
Comments and Suggestions for Authors
This is a well-written paper that seeks to compare hearing in noise related to detection of harmonic and inharmonic complex tones then look at frequency/amplitude modulation in noise in musicians and non-musicians.
Although well written, authors should explain a bit more why this work is important. The authors describe replication of previous studies, however they should explain why the further experimentation of frequency and amplitude modulation detection was undertaken, and why it would be important to know this. Is this important for hearing aid formulations? Would it be helpful in increasing understanding in a noisy environment?
The authors briefly discuss the neural substrates, however, further explanation might be helpful, for instance, why is this information not processed in the cochlea but in higher centers? The abstract mentions a general theory accounting for the effects of noise and memory on pitch-discrimination, which would be interesting, but this theory is not discussed in the manuscript. That would be very helpful.
Reviewer 2 Report
Comments and Suggestions for Authors
Introduction
1. Please add a rationale for the independent variable of musical training
2. It seems odd that the results are summarized at the end of the Introduction. Unless it is a requirement of the Journal, I would suggest removing this part (from line 77).
Methods
1. Masking noise - Please provide a rationale for using this noise as a masker for the experiments.
2. Detection in noise – Did the harmonic and inharmonic conditions repeat in blocks?
Results
1. Detection in noise – some outliers exist in the musicians and non-musicians groups. Were they the same participants for harmonic and inharmonic sounds?
2. Fundamental frequency discrimination – were the data log-transformed because of differences in variances? This, or any other reason, should be mentioned (only at the end of the paragraph individual differences are mentioned).
Discussion
The explanations are interesting and I am curious to see follow-up experiments testing them.
